# Study on Pelletizing Process of Spherical Activated Carbon Based on Molten Pitch

**DOI:** 10.3390/ma16072612

**Published:** 2023-03-25

**Authors:** Yaosen Wang, Hongsheng Qiu, Mo’men Ayasrah

**Affiliations:** 1School of Transportation and Logistics Engineering, Wuhan University of Technology, Wuhan 430063, China; 2Department of Civil Engineering, Faculty of Engineering, Al Al-Bayt University, Mafraq 25113, Jordan

**Keywords:** asphalt-based, spherical activated carbon, rotational viscosity experiment, thermogravimetric, prilling tower, melt asphalt

## Abstract

In the process of preparing asphalt-based spherical activated carbon, the molten asphalt must be formed into qualified spherical particles through the granulation process. Taking the process of molten asphalt granulation as the research direction, this paper carries out an asphalt rotational viscosity experiment and a thermogravimetric differential thermal experiment (TG–DSC), and obtains the optimal temperature and viscosity values for the asphalt granulation process. The fluent module in ANSYS software is used to input the known asphalt and prilling tower parameters. Based on the asphalt prilling principle, the thermal environment in the prilling tower during on-site melting and asphalt prilling is simulated. The results show that No. 70 matrix asphalt has good fluidity at 135 °C, and that, subsequently, the viscosity of the asphalt is stable and the fluidity of asphalt remains good with the increase in temperature; they also showed that the air velocity is fastest in the central area of the prilling tower, the air temperature is the highest at the top of the tower, and the air temperature in the central area is the lowest at the same height. Finally, a new approach to the granulation process of pitch-based spherical activated carbon is developed, which provides a reference for the basic experimental data and numerical simulation direction for the use of granulation towers to complete the granulation of molten asphalt in industry in the future.

## 1. Introduction

Asphalt-based spherical activated carbon is a spherical carbon material with asphalt as the main raw material, which is formed into spherical particles through unique processing technology, and then carbonized and activated at high temperatures in order to endow it with a high specific surface area, high adsorption performance, and high activity [1]. Asphalt-based spherical activated carbon is widely used in the military (armored motor vehicle and submarine life support systems, and response to biochemical war), environmental protection (purification of gas and liquid) [2], medicine (blood purification, drug treatment) [3], and chemistry (gas masks) [4]. It belongs to the top variety of the activated carbon family. In particular, its spherical characteristics are unmatched by other shapes of activated carbon. However, its complex production process and high technical cost are the main obstacles to the industrial production and promotion of asphalt-based spherical activated carbon. At present, the only global commodity supplier of asphalt-based spherical activated carbon is Japan’s KUREHA Chemical Co., Ltd. (Düsseldorf, Japan) [5]. Therefore, exploring the industrialized production path of asphalt-based spherical activated carbon is expected to break the Japanese monopoly and reduce the production cost.

Granulation towers are generally used in the chemical industry for cooling molten urea, ammonium nitrate, and other products. Through a process of spraying high-temperature molten liquid on the top of the tower and using air as the cooling medium, the granular products are collected at the bottom of the tower. However, the traditional emulsification pelletizing process is only applicable in the laboratory, and it is still impossible to mass-produce spherical asphalt industrially in in China. This paper explores the feasibility of using a granulating tower for molten asphalt granulation, by optimizing the structure of the traditional granulating tower, and provides basic data for the industrialization of the asphalt granulation process.

In addition, the melting device is an important factor in the granulation stage of an asphalt-based spherical activated carbon industrial production device. Its main function is to melt asphalt through high-temperature heating. In a laboratory environment, many domestic R&D units can produce qualified samples, but, so far, no domestic enterprise can engage in industrialized mass production of this product. Among many factors which restrict the industrialization of asphalt-based spherical activated carbon, the granulation process is key. In this paper, the granulation process of molten asphalt in a prilling tower is studied through a rotating viscosity experiment and a thermogravimetric differential thermal experiment. Combined with Ansys FLUENT software [6,7], the thermal environment in the prilling tower is simulated to explore the high tower granulation of molten asphalt, so as to lay a foundation for the industrialization of the asphalt granulation process in the future. Furthermore, we created a new method of asphalt granulation using the granulation tower.

Through the simulation of the internal thermal environment of the granulation tower, this paper closely studied the feasibility of the granulation of molten asphalt through the granulation tower, analyzed the air temperature field and air velocity field in the tower, and studied the heat exchange between the molten asphalt and the air in the tower, providing basic data and references for the subsequent modification and optimization of the granulation tower and improving the granulation quality. In this paper, the movement process of molten asphalt particles with different particle sizes in the granulation tower is studied in detail, and the movement of asphalt particles in the granulation tower is divided into three stages, namely: 1. the liquid cooling stage; 2. the solidification stage; 3. a solid cooling stage. The conclusions obtained are helpful for the production of asphalt particles that meet the particle size requirements of an industrial granulation tower.

## 2. Materials and Methods

### 2.1. Materials

The production process of asphalt-based spherical activated carbon has three parts: the first part is asphalt balling [6], the second part is nonmelting treatment, and the third part is carbonization and activation. High-softening-point asphalt, with a softening point higher than 250 °C, is generally used as the raw material for asphalt spheroidization, and is thus subject to subsequent nonmelting treatment. Asphalt is a hot-melt material, and the spherical particles still have the characteristics of phase transformation in the case of hot melting, so they also need nonmelting treatment (also known as pre oxidation treatment); that is, the resin components in asphalt are cross-linked by high-temperature oxidation to become a solid, nonmelting, and insoluble substance. This process has high requirements for the raw asphalt material used [7]; it can only be carried out with high-softening-point asphalt. However, high-softening-point asphalt is very expensive, and the research focus of this paper is molten asphalt granulation, which does not involve the subsequent nonmelting treatment of asphalt. Therefore, ordinary No. 70 matrix asphalt is used for the experiment, and its performance indexes are shown in Table 1.

Using the high granulation tower structure, the asphalt is melted at the bottom of the tower first, and then the molten asphalt is transported to the rotary granulation nozzle on the top of the tower through the pipeline and sprayed into the granulation tower. In the process of transporting molten asphalt from the bottom of the tower to the top of the tower, due to the long transportation distance (the tower height is greater than 100 m), the asphalt temperature needs to be strictly controlled. If the temperature is too low, the asphalt is likely to lose fluidity in the pipeline and block it. If the asphalt temperature is too high, it is likely to decompose, causing its physical properties to change, and making it impossible to produce spherical asphalt that meets the industrial requirements. The working principle of an asphalt granulation tower is to transport the molten asphalt to the rotary granulation nozzle on the top of the tower and spray it to form a high-temperature jet. Once sprayed, each particle falls along the same paraboloid and forms a spiral line. In the process of falling, the molten asphalt jet breaks quickly into droplets [8]. If the viscosity of asphalt is too high, it cannot break into small droplets during spraying, which will affect the particle size and quality of the final product. In view of the above two situations, with viscosity as the index, a rotational viscosity experiment is carried out to judge whether the fluidity of molten asphalt at high temperatures meets the requirements. The higher the viscosity of the asphalt, the lower the fluidity. Furthermore, a thermogravimetric differential thermal experiment (TG–DSC) was carried out to explore the effect of temperature on the physical properties of asphalt.

### 2.2. Rotational Viscosity and Thermogravimetric Differential Thermal Experiment

The U.S. SHRP strategic plan uses viscosity as an important index to evaluate asphalt pavement performance and asphalt temperature sensitivity. In the test, a 27 # rotor is selected, and the asphalt, added with naphthalene, is kept at 95 °C in the viscosity meter for 30 min before the test is carried out. A high speed is selected, and the torque range is controlled between 5% and 95%. Once the viscosity is relatively stable, data are recorded, before the temperature is raised to 135 °C, and the trend of the change in viscosity of naphthalene-containing asphalt is observed.

Thermal analysis is the use of programs to control temperature, measure the physical properties of substances, and determine the relationship between temperature and time. Differential scanning calorimetry (DSC) is a technology used to measure the difference between the ability of the sample and the reference material, and the temperature relationship under the program-controlled temperature. The predecessor of this technology is differential thermal analysis (DTA), but the former is superior to DTA in quantitative analysis. The abscissa of the DSC curve is the temperature (°C), and the ordinate is the heat flux (dH/dt). The displacement of the curve from the baseline represents the rate of heat absorption, or release, of the sample (mJ/s).

### 2.3. Experimental Result

One of the key characteristics of asphalt-based spherical activated carbon is its huge internal porosity, i.e., specific surface area. In addition to the porous characteristics of carbon materials, organic solvents need to be added to asphalt to promote pore formation, and organic solvents can also improve the fluidity of molten asphalt [9]. Common organic solvents include benzene (melting point 5.5 °C, boiling point 80.1 °C), naphthalene (melting point 80.5 °C, boiling point 217.9 °C), and anthracene (melting point 218 °C, boiling point 342 °C). Because the melting point of naphthalene is close to the softening point of No. 70 base asphalt, the temperature of the molten asphalt is always lower than the boiling point of naphthalene during the heating process, and naphthalene will not vaporize or sublimate, which greatly reduces the difficulty of adding solvent to the molten asphalt. Therefore, naphthalene is used as a pore forming agent in this paper. No. 70 matrix asphalt and naphthalene are placed in a heating box and heated to 170 °C and then removed. No. 70 base asphalt is divided into three parts, with 10%, 20% and 30% naphthalene added, respectively; these are stirred fully and evenly [10], the sample is prepared, and the rotational viscosity experiment is conducted. The experimental results are shown in Figure 1.

It can be seen from Figure 1 that, with the increase in temperature, the viscosity of the asphalt sample decreases and the fluidity increases. When the temperature reaches approx. 135 °C, the slope of the curve approaches 0. If heating continues, the viscosity of the asphalt changes little, and its fluidity remains unchanged. When the content of naphthalene is 30%, the viscosity of asphalt is the lowest and the fluidity is the best. Therefore, the 30% content of naphthalene is selected. The related papers show that the process of extracting naphthalene from spherical particles with solvent is also easier than that of benzene and anthracene, which reduces the difficulty of the subsequent process.

The green curve in Figure 2 is the TG curve of No. 70 base asphalt. It can be seen that the TG curve is a smooth inverse S curve. The curve shows that the asphalt begins to decompose at about 280 °C, after which the asphalt quality decreases sharply. Below 280 °C, there is no change in the quality of asphalt that can be detected by an instrument [11]. In the temperature range from 0 °C to 100 °C, the asphalt content exceeds 100%, which is due to the poor air-tightness of the instrument. Mixing in some air leads to the oxidation reaction of asphalt at this temperature, resulting in an increase in asphalt quality.

With the rise of temperature, the phase state of asphalt will change. In the DSC curve, there is a peak. However, as asphalt is a mixture, the phase state transition temperature between the different components is different, and there are overlapping peaks in the DSC curve. Therefore, the DSC curve of asphalt is not as obvious as that of other materials, and there is no certain transition temperature [12]. Nevertheless, two important pieces of information can be obtained from the DSC curve of asphalt; one is the size of the enthalpy change, or the area of the endothermic peak, and the other is the location of the endothermic peak, or the temperature range of the endothermic peak. Both reflect the aggregation state of asphalt within the temperature range in which the endothermic peak occurs. The size and position of the peak reflect the micro properties of asphalt. The larger the peak, the larger the area surrounded by the peak, indicating that there are many components of asphalt that change in this temperature range, greatly affecting the physical properties of asphalt. After heating, the physical properties of asphalt also change greatly, which indicates that the thermal stability of asphalt is poor.

The blue curve in Figure 2 is the DSC curve of No. 70 base asphalt. It can be seen that the first peak of asphalt appears between 120 °C and 200 °C, which is endothermic, but that this peak is very small compared with the later peak, indicating that there are few components that change in this temperature range. It can be assumed that the physical properties of asphalt essentially do not change. When the temperature exceeds 300 °C, two large endothermic peaks appear, indicating that the physical properties of asphalt have changed greatly within this temperature range, resulting in thermal decomposition.

Because the asphalt is in a flowing state at high temperatures, its physical and chemical properties change little below 400 °C, and the fluidity of the asphalt is further increased after the addition of naphthalene, it is feasible to granulate the molten asphalt through the granulation tower.

According to the comprehensive analysis in Figure 1 and Figure 2, considering the temperature loss and fluidity reduction of asphalt transported from the tower bottom to the tower top, and in order to prevent the pipeline from being blocked by cooling asphalt, the temperature of the molten asphalt is set at 170 °C; the temperature loss after reaching the tower top is 30 °C. At this temperature, naphthalene and asphalt are in a flowing state, which is convenient for the complete mixing of the two to form holes.

## 3. Numerical Simulation

### 3.1. Numerical Model

A high pelletizing tower structure is used in the model, and the structure is shown in Figure 3. The molten asphalt transported from the tower bottom is sprayed by the rotary granulation nozzle, and air is used as the cooling medium [13]. As the asphalt is falling, under its own weight, it is cooled and solidified by the rising air flow [14], and spherical asphalt particles are collected at the tower bottom [15].

In the granulation process, the height of the granulation tower must meet the heat exchange time required for the liquid droplets of molten asphalt to change from liquid phase to solid phase, and the droplets cannot come into contact with the inner wall of the granulation tower while falling, otherwise they will stick to it [16]. After calculation, the height of the granulation tower is set at 120 m, the diameter is set at 30 m, 12 air inlets are located at the bottom of the tower, the height of the air inlets is 3 m, the axis of the granulation tower is asymmetric [17], and the air outlet is located at the top of the tower [18,19,20]. As the molten asphalt is ejected from the top of the tower and falls to the bottom of the tower, air enters from the air inlet at the bottom of the granulation tower due to the action of hot air pressure, since the temperature of the tower top is higher than that of the tower bottom [21]. After the asphalt particles falling from the top of the tower are heated, the air temperature rises, and the air is discharged from the upper air outlet to complete the heat exchange [22].

For the granulation tower model grid, a combination of structured and unstructured methods are adopted, the boundary conditions are set, physical parameters added, the air inlet and outlet set as pressure inlet and pressure outlet, and the air inlet temperature set at 297 K (24 °C), according to the field measured data [23]. The temperature of the molten asphalt is 170 °C, while the mass flow is 239.54 t/h, and the asphalt is expelled from the rotary granulation nozzle with a rotating speed of 30.55 rad/s. The simulation adopts the DPM in full model, the discrete method adopts the finite difference method of the control volume, the continuous phase equation is solved iteratively by a simple algorithm, and the dispersed phase is calculated by steady-state coupling [24]. The latent heat of the asphalt particles is approximated by modifying the specific heat capacity, and the natural convection of air in the granulation tower is approximated by Boussinesq [25].

### 3.2. Simulation Result

As the melted asphalt is ejected from the nozzle, there is an initial velocity in the horizontal direction, which is determined by the rotation speed of the nozzle. In the vertical direction, it falls under the action of gravity and air resistance. The ejected molten droplet makes a para-curve motion in the granulation tower. In the actual calculation, the droplet motion can be drawn into a plane problem. Because the granulation tower is closed around, the air resistance in the horizontal direction can be ignored.

According to the cloud diagram of air temperature in the granulation tower in Figure 4 and Figure 5, the temperature contour presents an irregular W-shape (the air inlet is asymmetric, so it presents the image of low on the left and high on the right). There is a situation of highs in the middle and lows on both sides. The specific reason is that the rotating granulation nozzle is installed in the middle of the tower top, and the molten asphalt is sprayed evenly from the nozzle, most being concentrated in a ring 5–9 m away from the center of the tower. The rising air in this area has a heat exchange with more molten asphalt particles, and the air temperature rises greatly. The air directly below the rotary granulation nozzle and around the ring has less contact with the molten asphalt, and the temperature is lower. The air temperature in the tower increases with the increasing height of the granulation tower. When the air reaches the air outlet at the top of the tower, its temperature has increased by nearly 30 °C compared with that at the bottom of the tower.

The average air temperature of the cross section of the granulation tower at this height was taken every 10 m, as shown in Figure 6. It can be seen from Figure 6 that the air temperature in the tower increases with the increase in the height of the tower, and the slope of the curve increases, indicating that the change rate of air temperature is faster closer to the tower top, and most of the heat exchange between asphalt and air is completed above the granulation tower.

The cross section of the central axis was taken with the air inlet of the granulation tower as the origin; the upward direction was positive, and four cross sections were taken for air temperature analysis, which are, respectively, set at Z1 = 20.4 m, Z2 = 42.1 m, Z3 = 58.7 m, and Z4 = 70.4 m. The curve is shown in Figure 7. Due to the fact that most particles are concentrated in an area 6 to 11 m from the radius of the granulation tower, the heat exchange between asphalt particles and air in this area is more frequent than in other areas, and the air temperature is relatively higher, showing a peak on the curve. At this distance from the center of the tower, due to the sparse particles in the area and the minimal heat exchange between the air and particles, the temperature in the same cross section is low, showing the shape of a valley. At a location near the wall of the granulation tower, due to the sparse distribution of asphalt particles as compared to the center of the granulation tower, the heat exchange between the air and asphalt particles is also small, and the air temperature is low. Therefore, the radial distribution curve of the cross section of the air temperature is generally M-shaped. The M shape is not strictly symmetrical about the axis of the granulation tower, which may be caused by the incomplete symmetry of the internal structure of the granulation tower and the air inlet and outlet.

As can be seen from Figure 8, the wind speed is the largest at the air inlet, and the speed direction is horizontal. After entering the granulation tower, the air is affected by the thermal pressure on the top of the tower, and the speed direction changes from horizontal to vertical. The speed decreases significantly and tends to be stable. The air flow velocity in the granulation tower is between 0.6–0.8 m/s. Flow can be found as
(1)Q=v∗S
where v is the air flow rate, in meters per second, S is the cross-sectional area, in square meters, Q is the air flow rate, in cubic meters per second. When the air enters the granulation tower from the air inlet, the air velocity decreases, because the cross-sectional area of the granulation tower is greater than that of the air inlet. In the central area of the granulation tower the number of asphalt particles is low, the resistance of the air in the rising process is small, and the air flow rate is the fastest. In the area 5–9 m away from the center of the tower, the air collides with a large number of falling asphalt particles during the rising process, increasing the resistance and reducing the air flow rate. At the periphery of the granulation tower, due to fewer asphalt particles, the resistance of the air to the asphalt particles is small, and the air flow rate is large.

The average air velocity data was taken from several different cross-sectional heights and placed directly below the air outlet, then the curve was drawn. It can be seen from Figure 9 that, after the air enters the granulation tower from the air inlet, the velocity decreases sharply, and the air velocity then changes little in the height range of 20–100 m. When the air rises close to the air outlet on the top of the tower, the air velocity increases rapidly. This shows that the air flow rate in the granulation tower is mainly affected by two factors: first, the change of cross-sectional area will lead to a change in the air flow rate. The smaller the cross section of air passing through, the greater the air flow rate. Second, the more intense the heat exchange between the asphalt particles and the air, the greater the impact on the air flow rate. However, the air flow rate is at a lower level in the area of 20–100 m in the granulation tower. This is because the asphalt particles will also have a certain resistance to the rise of the air as it falls in the granulation tower. Only when the heat exchange between the asphalt particles and the air reaches a certain degree, when approaching the tower top, will the air speed increase significantly.

As shown in Figure 10, most particles are concentrated at a distance of 6~11 m from the radius of the granulation tower, which will not only lead to frequent heat exchange between the air and particles in this area, resulting in an increase in air temperature, but also lead to a lower average air flow rate in this area, due to the greater resistance of a large number of particles against the air, than in other areas, showing a valley shape on the curve. Since there is minimal particle distribution in the center of the granulation tower and near the tower wall, the air in these two areas is subject to little rising resistance, so the air flow rate is higher, and the “peak” of the curve is concentrated near the center of the granulation tower. Therefore, the radial distribution curve of the cross section of the air flow rate is opposite to the radial distribution curve of the air temperature, showing a W shape. The curve for a cross section height of 20.4 m has multiple peaks, and the shape of the curve is somewhat irregular, which may be caused by the air turbulence, as the cross section is close to the air inlet.

The particle size of the pitch-based spherical activated carbon used in most fields is mostly between 1 mm and 3 mm. Therefore, from the perspective of economic benefits, whether the granulating tower can produce qualified pitch spheres within this particle size range or not is crucial. Therefore, this paper selects asphalt particles with particle sizes of 1.0 mm, 1.5 mm and 2.5 mm in order to track their movement process in the granulation tower, uses the Fluent finite element model to calculate the change of particle temperature of asphalt particles in the granulation tower with drop distance, and extracts data to draw a curve, as shown in Figure 11.

The particles fall into the granulation tower and undergo convective heat exchange with the rising air. The temperature of the particles gradually decreases, and they then go through the liquid cooling process, solidification process, and solid cooling process. From Figure 10, it can be seen that those particles with a particle size of 1 mm must drop by about 10 m to enter into the solidification process, and drop by about 40 m to enter the solid cooling process, with a final particle temperature of 50 °C; the 1.5 mm particle size drops by about 15 m to enter the solidification process, and drops by about 50 m to enter the solid cooling process, with the final particle temperature is 63 °C; however, the 2.5 mm particle size drops by about 30 m and enters the solidification state. When it falls to the bottom of the tower, it is still in the solidification state, and the final particle temperature reaches 105 °C. The difference between the heat transfer coefficient and residence time of particles with different particle sizes finally leads to the difference between the particle cooling curves. The convective heat transfer coefficient and residence time of particles with large particle sizes are smaller than those of particles with small particle sizes. The latent heat of solidification has not been completely released whilst falling to the bottom of the tower.

As shown in Figure 12, when the molten asphalt is sprayed out, the initial vertical falling speed is 0 m/s. However, the vertical gravity of the particles is greater than the vertical air resistance, and the falling speed of the asphalt particles gradually increases. With the increase in the speed of the asphalt particles, the air resistance also increases, and the vertical acceleration of the asphalt gradually decreases to zero. Of the selected sizes, 2.5 mm asphalt particles begin to fall at a constant speed after falling for about 30 m, and these particles complete most of their journey in the granulation tower at a constant speed. From Figure 12, it can be seen that the larger the particle size of asphalt particles, the smaller the air resistance relative to its gravity, and the greater the maximum falling speed it can reach. The smaller the particle size of asphalt particles, the greater the impact of air resistance, and the smaller the maximum speed that can be reached. This means it takes longer for particles with small particle sizes to reach the bottom of the tower, and it takes a shorter time for particles with large particle sizes to reach the bottom of the tower, due to their higher vertical falling speed. Additionally, when a particle is close to the air inlet, it will be more disturbed by the incoming air. Therefore, it can be seen from Figure 12 that the 1 mm asphalt particles begin to fall at a constant speed after falling about 15 m, and the 1.5 mm asphalt particles begin to fall at a constant speed after falling about 20 m.

## 4. Conclusions

Through the experimental study on the thermal behavior of No. 70 matrix asphalt, and the fluid simulation of the asphalt granulation process, the optimal temperature of asphalt granulation and the distribution of air temperature and velocity in the granulation tower are obtained. The conclusions are as follows:Controlling the temperature of the molten asphalt at 170 °C is conducive to the full integration of asphalt and the pore forming agent. At this temperature, the asphalt has good fluidity and is not prone to pipeline blockage. At the same time, at this temperature, the components of asphalt are stable, and the physical properties are essentially unchanged.The average air temperature at the bottom of the granulation tower is the lowest. With the increase in the height of the granulation tower, the average air temperature increases, and the average air temperature is the highest at the top of the tower. At the same cross-sectional height, the air temperature presents the characteristics of “two highs and three lows”, which are reflected in an irregular W-shape on the temperature cloud diagram.After the air enters the granulation tower from the air inlet, the air velocity decreases. The air velocity in the central area of the granulation tower is the highest, the air velocity at 5–9 m from the center is the lowest, and the air velocity at the outer edge of the granulation tower is significant. With the increase in the height of the granulation tower, the average air velocity becomes lower and lower. When the air reaches the air outlet on the top of the tower, the air velocity increases sharply.According to the cross-sectional air temperature cloud diagram, the coverage of molten asphalt particles in the granulation tower is small, resulting in a certain waste of production capacity. Under production conditions that meet the environmental protection requirements, the spraying amount of the granulation nozzle can be appropriately increased, the nozzle speed can be increased, the coverage of molten asphalt particles in the granulation tower can be improved, and the output can be increased.The temperature of 2.5 mm asphalt particles falling to the bottom of the tower is too high. At this temperature, the contact with the bottom of the granulation tower is liable to cause deformation, which does not meet the production requirements.From the analysis of the numerical simulation results, the granulation tower cannot produce spherical asphalt particles with a particle size greater than 2.5 mm. The cooling effect of the granulation tower can be strengthened by adding an air inlet and outlet to the tower to increase the ventilation rate. The temperature of asphalt particles can also be reduced by properly reducing the discharge temperature of the molten asphalt.

## Figures and Tables

**Figure 1 materials-16-02612-f001:**
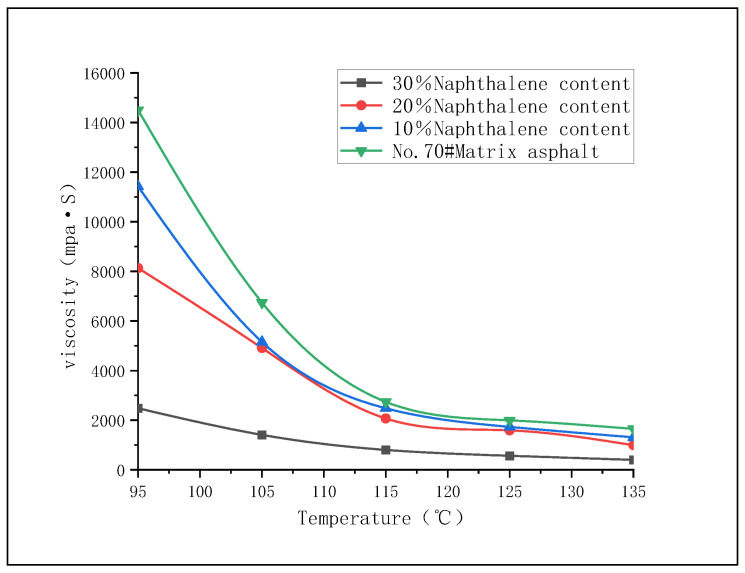
Viscosity curve of asphalt with different naphthalene content.

**Figure 2 materials-16-02612-f002:**
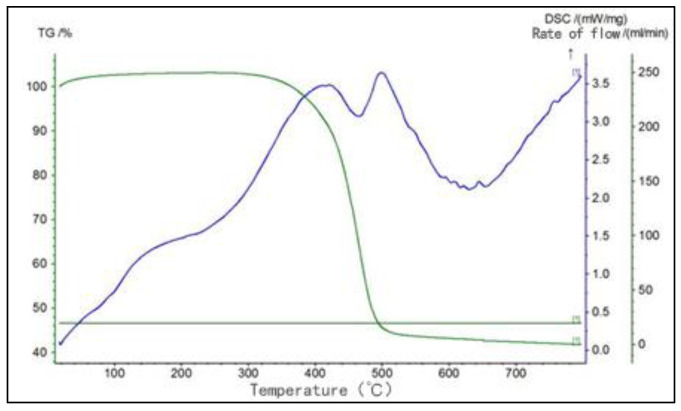
TG–DSC curve of 70 base asphalt.

**Figure 3 materials-16-02612-f003:**
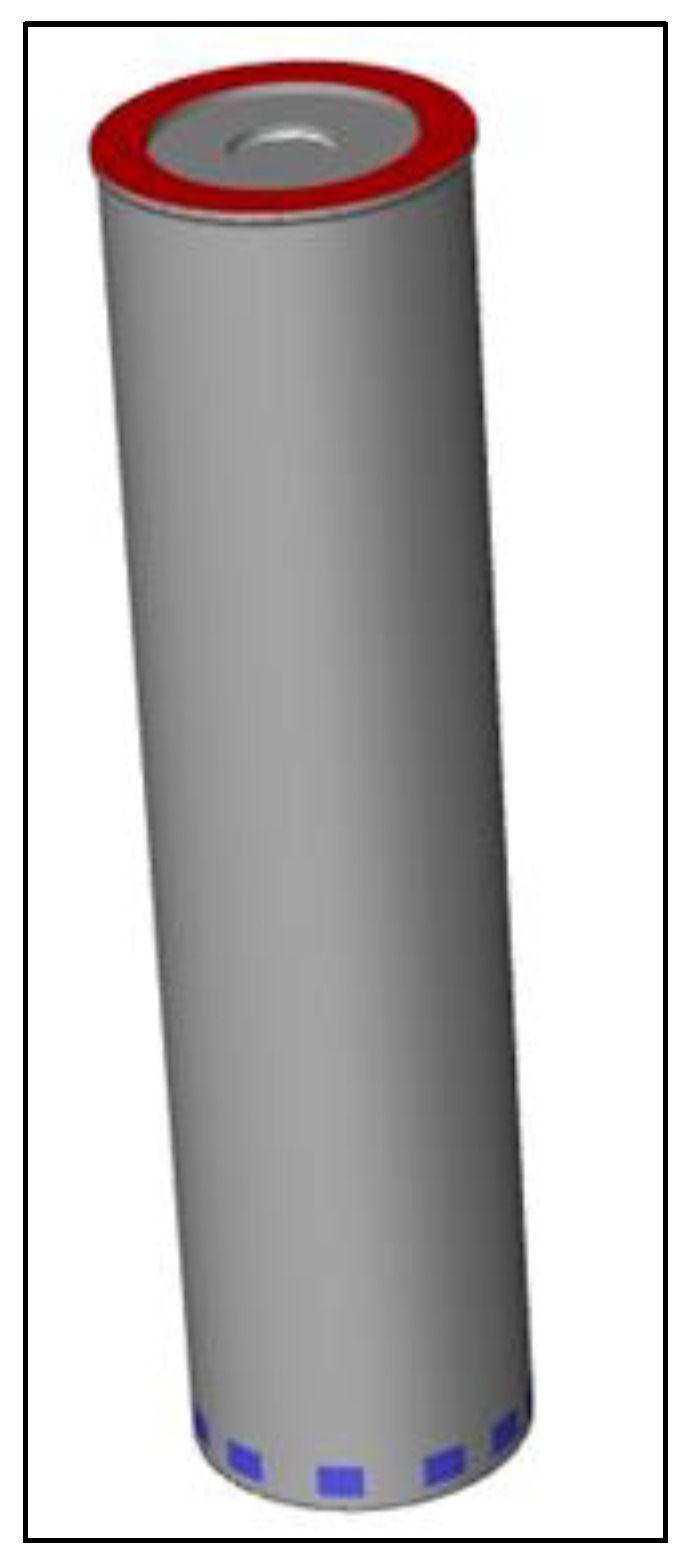
Prilling tower model diagram.

**Figure 4 materials-16-02612-f004:**
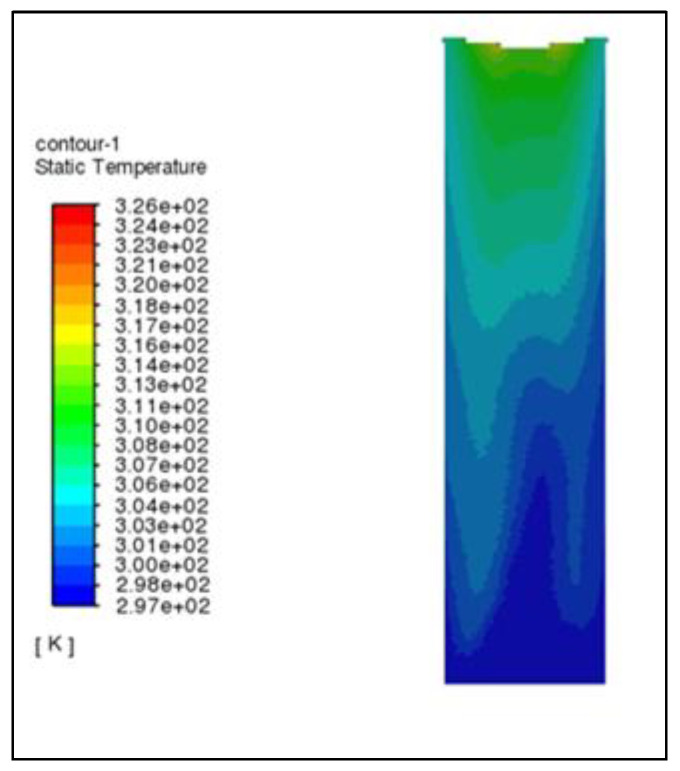
Air temperature nephogram.

**Figure 5 materials-16-02612-f005:**
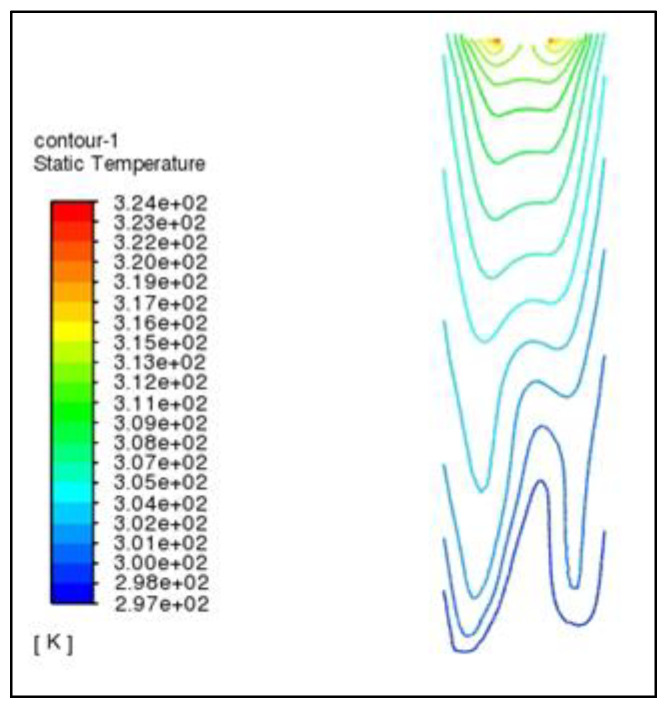
Air isotherm cloud chart.

**Figure 6 materials-16-02612-f006:**
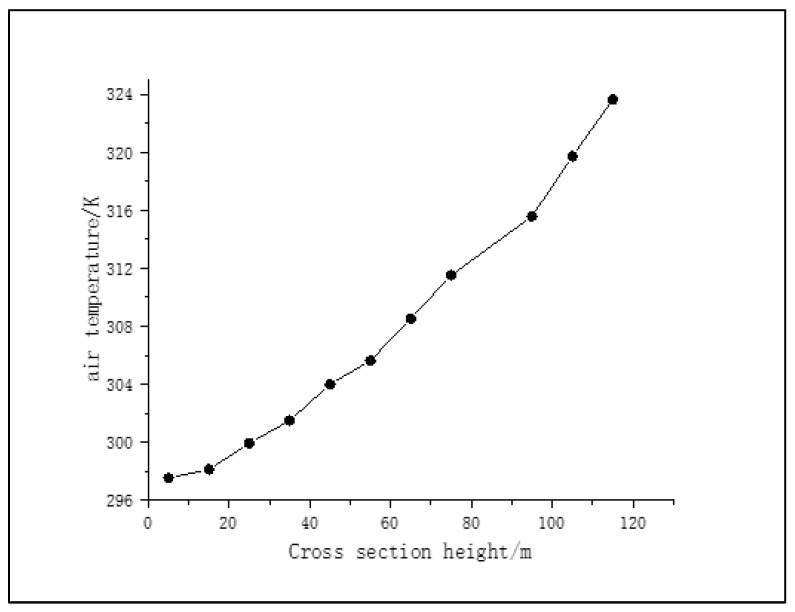
Average air temperature at different cross section heights.

**Figure 7 materials-16-02612-f007:**
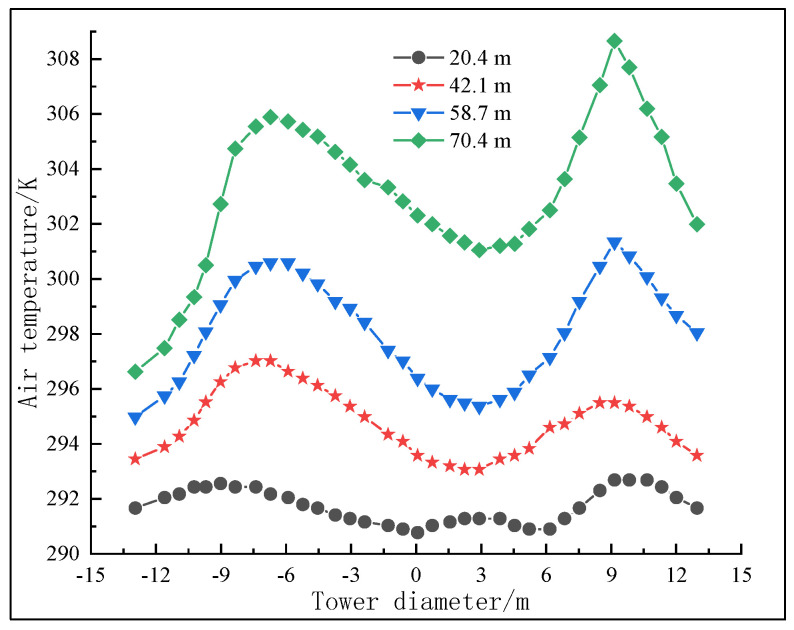
Radial distribution diagram of temperature at cross section of granulation tower.

**Figure 8 materials-16-02612-f008:**
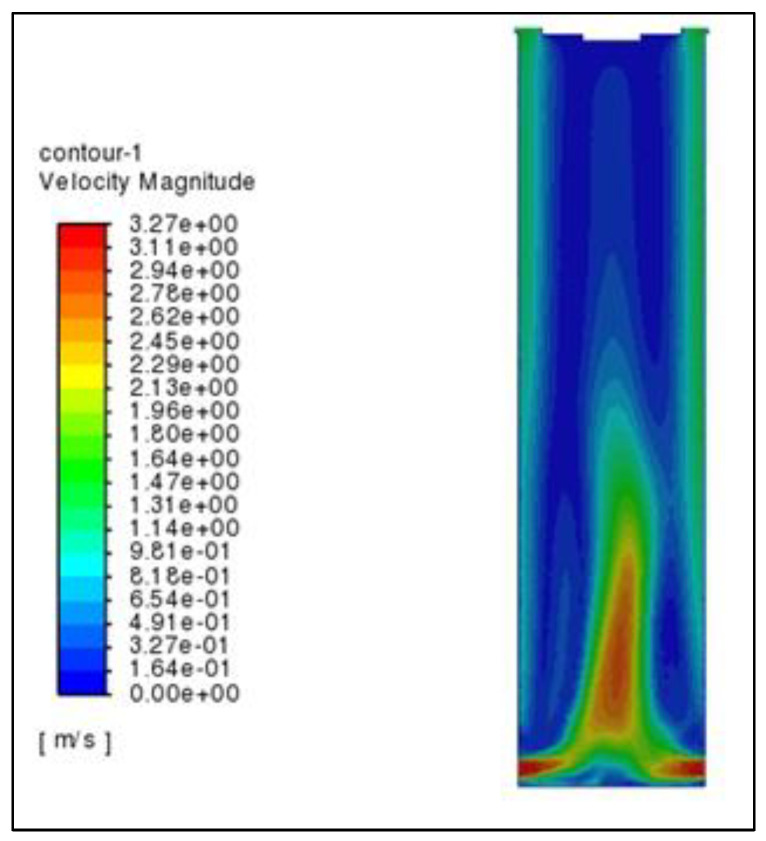
Air velocity nephogram.

**Figure 9 materials-16-02612-f009:**
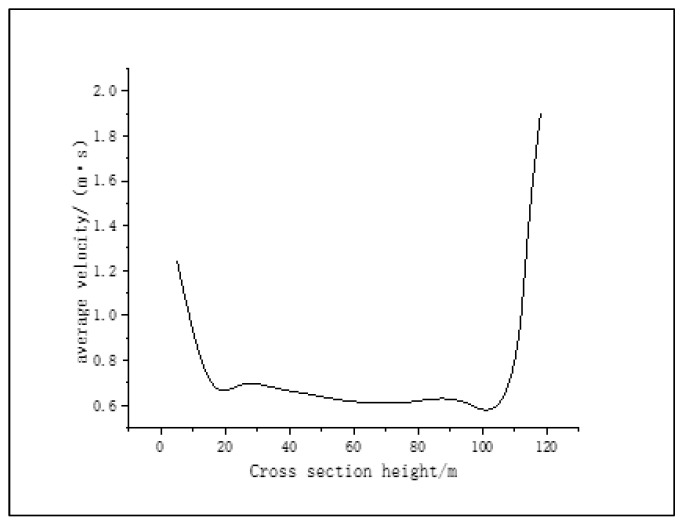
Average air velocity at different cross-sectional heights.

**Figure 10 materials-16-02612-f010:**
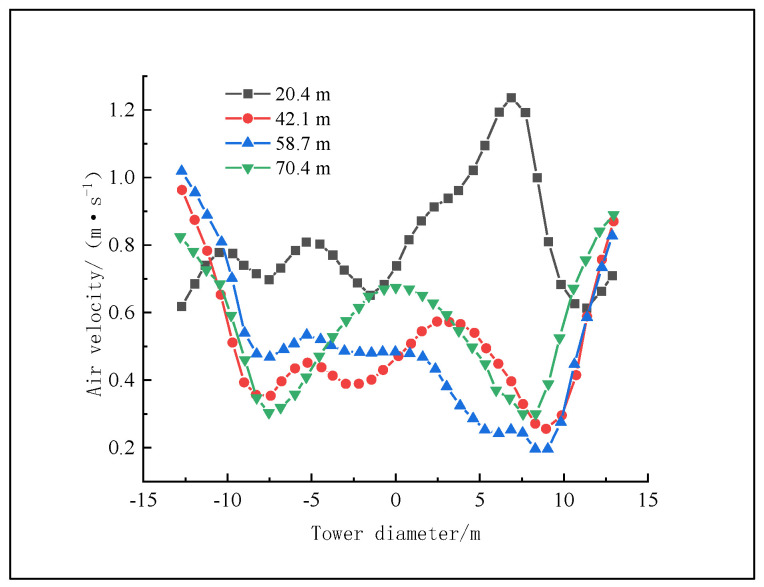
Radial distribution of air velocity in the cross section of the granulation tower.

**Figure 11 materials-16-02612-f011:**
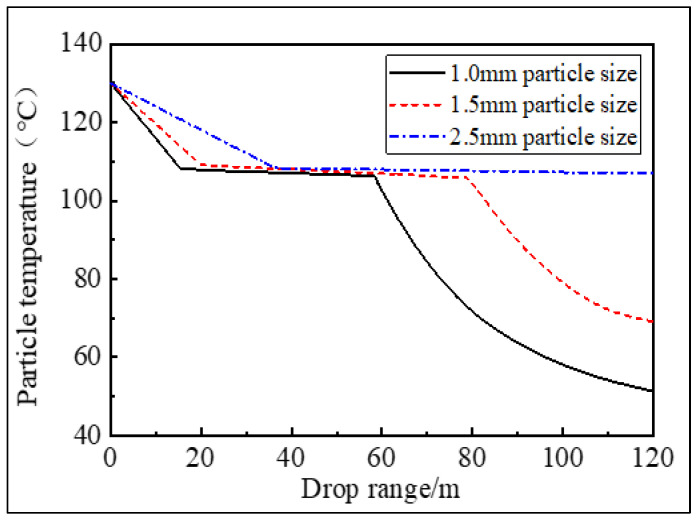
The diagram of particle cooling in tower.

**Figure 12 materials-16-02612-f012:**
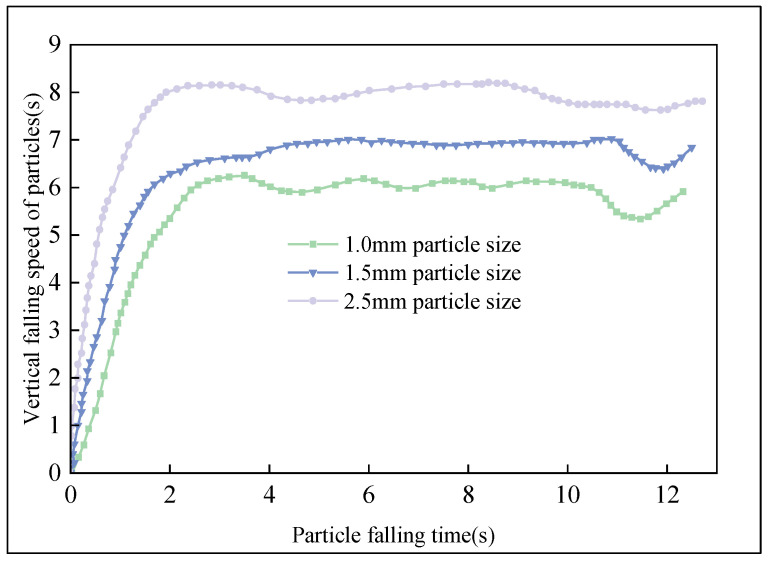
Vertical velocity variation of particles in the tower.

**Table 1 materials-16-02612-t001:** 70#Asphalt performance index table.

Project	Quality Index	Measured Index
Penetration (25 °C, 100 g, 5 S) (0.1 mm)	60–80	72.6
Softening point (°C) not less than	46	47.9
Ductility (5 cm/min) not less than (cm)	20 (10 °C)	63.91 (°C)

## Data Availability

The data presented in this study are available on request from the corresponding authors.

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
