# Peer review of "Study on Pelletizing Process of Spherical Activated Carbon Based on Molten Pitch"

_materials, 2023, doi:10.3390/ma16072612_

Round 1

Reviewer 1 Report

This article discussed on the asphalt granulation process by conducting the rotational viscoity test and thermal test (TG-DSC). Some comments to be address by the authors are listed below:

Title : The title does not accurately reflect the article's content. Suggest to change the title to something more relevant to the content. (Use the terms molten asphalt or spherical activated carbon based on asphalt)

Line 51: R & D in full

Line 57: typo experiment, Combine

Subsection 2.2 and 2.3: This section should discuss the methods used. The results on rotational viscosity and TG-DSC should be written in Subsection 3.

Line 151: Typo peak. but

Suggest the authors to cite relevant paper to support the results for rotational viscosity and TG-DSC

Figure 4, 5, 6, 7 and 8 sholuld be placed after explanation of each figure.

Line 208 : DPM in full

Line 253: Flow Q = v x S should be written as Equation. Not embedded in the paragraph.

Figure 9 : y- axis (Particle) 

Line 273-278: Why are the particle sizes chosen to be 1mm, 1.5mm, and 2.5mm? Is there a reference?

Author Response

The authors would like to thank the Reviewer for his worthy comments because they will make the manuscript valuable.

Reviewer 2 Report

Dear Authors,

I am happy to share that your work has been presented well, but the only concern from my side is related to no. of graph plots in the manuscript. It is suggested that authors should add some more no. plots in the result and discussion section for proper clarity of conclusion. And also show some comparative study graphs for validation of your work.

Author Response

(The authors gave the same response as above.)

Reviewer 3 Report

Manuscript; materials-2296871

Article title: Thermal Behavior of Asphalt and Simulation of Melt Granula-tion Environment                       

The problem is timely and interesting. I recommend the publication of the manuscript after the following revisions are properly made:

1) The aim of the paper is not completely well-specified. The authors could specify more this aspect in the abstract and in the introduction of the manuscript.

2) Please, check if all the symbols used in the various equations are well-defined in the text.

3) English should be enhanced throughout the manuscript to eliminate grammatical errors and misprints.

4) In this paper ANSYS software is used. For general readers, authors are encouraged to discuss other kind of works on FEM such as: [(a) “Microstructural/geometric imperfection sensitivity on the vibration response of geometrically discontinuous bi-directional functionally graded plates (2D-FGPs) with partial supports by using FEM”, Steel and Composite Structures, 45(5), 621-640.; (b) “Static bending and buckling analysis of bi-directional functionally graded porous plates using an improved first-order shear deformation theory and FEM”, European Journal of Mechanics - A/Solids, 96, 104743.].

5) Figs. 7 and 8 should be more discussed.

6) In conclusion, give only main findings of your research with an appropriate value.

Author Response

(The authors gave the same response as above.)
